# Image Embeddings from Social Media: Computer Vision and Human in the Loop Applications for Social Movement Messaging

## Abstract

Using images to understand messaging systems has been done qualitatively for smaller batches, particularly in the news context, but less has been done from a quantitative approach on domain specific images. To review the messaging structure of a larger number of topical images, 16,567 image posts from Instagram related to the anti-feminicide movement in Mexico were collected and analyzed. The analysis included using unsupervised (ResNet50) and self-supervised computer vision models (CLIP, BLIP-2's embedding model) on the image feature embeddings, and evaluated with tuned density based metrics (HDBSCAN). Human in the loop evaluation is also applied through a content analysis of top images within each cluster to compare the various facets of representation within the image collection. Clustering shows that the embeddings are densely packed, representing visual overlap across the collection of topical images. Human in the loop content analysis enabled a closer reading of the visual images, especially those that contained text which found a range of the topics including, woman/male comparisons, accusers, life examples, domestic violence, gender violence, protest phrasing, support or solidarity of someone or a cause. The comparison showed that the best separation results came from the CLIP model, but still shows a lot of overlap.

## 1 Introduction

Social media images are never truly stand-alone, they are grouped to share a specific message sprinkled across a user's feed. Understanding the message of these groups is increasingly important as more people get their information from social media, especially about critical news topics. Using images to understand messaging systems has been done qualitatively for smaller batches, particularly in the news context (Ribeiro et al., 2018; Weeks et al., 2019), but less has been done from a quantitative approach on domain specific images (Liu & Panagiotakos, 2022). This work uses computer vision methods and Human in the Loop content analysis to understand the groupings of a collection of images with densely clustered features. Specifically, the work begins with the image itself as the unit of analysis and how similar its components are to the other images in the corpus by using feature embedding models and clustering those embeddings to identify common groups.

This uses 16,567 collected and cleaned Instagram posts of the anti-feminicide movement in Mexico, to review the messaging structure of the dataset as a whole. The work analyzes how similar an image's contents relate to the other images in the image data by comparing feature embedding model outputs from ResNet50, CLIP, and BLIP-2's. The embeddings are clustered to identify visually semantic groups. These grouped clusters are then analyzed using inductive content analysis to organize the data into various categories, concepts and themes. This evaluation is then compared to the quantitative evaluation metrics which strengthens the overall understanding of the visual messaging that comes from a topical group of Instagram posts.

### 1.1 Pattern Detection within Image Embeddings

As images have been referred in different ways by various disciplines (Mitchell, 1984), the social media image derives from a fragmented flow of information that comes with qualities of visual evidence, bias, and communication practices. The unit of analysis for this work is an image, more

specifically, the Instagram image.It's the image itself including all the attributes that accompany it, within the frame of the social media platform. This includes the attention it gets, the account that produced it, and the actual content of the image (Rose, 2022). In this research, an image is also analyzed relative to other, similar images. The assumption for analyzing an image in the context of similar images is that the whole will provide an environment for a larger narrative to represent a messaging structure of a community online.

The pattern detection analysis of this research solely focuses on the content of an image, including the linguistic meaning of the text that is found within many of the posts. Here, just looking at the image itself enables an analysis of Instagram's fragmented flow of visual information and represents the social movement's messaging as a standalone visual perception. Exploring the collection of images by grouping these messages together to identify patterns, supports the foundation for developing the meaning of the messaging being circulated by the anti-feminicide movement in Mexico.

## 2 CLUSTERING

This first step uses clustering, where images are grouped into their respective themes based on their content using foundational computer vision derived feature embeddings from ResNet50 and CLIP (He et al., 2015; Radford et al., 2021). The embeddings are clustered and evaluated to identify patterns brought about by each model with some rounds of parameter tuning. The image features are analyzed for how similar they are to each other using cosine similarity and minimum cluster sizes to tune the embedding model and get an initial idea of a good number of groups (Campello et al., 2013) while running methods like the Hierarchical Density-Based Spatial Clustering of Applications with Noise (HDBSCAN). Evaluation metrics like silhouette score, Calinski-Harabosz index, and human in the loop reviews are then used to identify clustering validity.

### 2.1 EMBEDDING EXTRACTION

To get feature embeddings, there was still some more cleaning to do as the first few runs of the images returned an error. This showed that some of the images downloaded were blocked and had no image at all. These images were identified and filtered out. Then the working images are preprocessed to be resized to all have the same image pixelation (224, 224), and then normalized for an RBG mean of (0.485, 0.456, 0.406) and an RBG standard deviation of (0.229, 0.224, 0.225). Three different feature extraction models were used on the pre-processed images, ResNet50, CLIP, and the embedding model for BLIP-2.

**ResNet50**: Because of its fixed feature size architecture of 50 classical convoluted neural network (CNN) layers that prove solid for numerous feature embedding performances (Radford et al., 2021), ResNet50 is chosen as a solid baseline for comparing the following CLIP and BLIP2 embedding applications. **CLIP**: contrastive language-image pretraining, uses a Vision Transformer (ViT) model pre-trained on about 400 million contrastive image text pairs to encode the images. The CLIP model in this work comes from HuggingFace's OpenAI and uses these image-text pairs for feature embedding and image classification tasks further down the pipeline. **BLIP-2**: Bootstrapping Language-Image Pre-training, is typically implemented for unified vision-language understanding and generation. Here, only BLIP-2's vision encoder is used in this part of analysis. The unimodal vision encoder divides the image into patches and encodes them as a sequence of embeddings (Li et al., 2023). This encoder is frozen and then uses a Querying Transformer (Q-Former) as another trainable module. The Q-former is a small transformer model that uses learnable query vectors on the frozen image encode (Li et al., 2023). Unlike CLIP, this model's vision encoder does not use image-text pairs, and is expected to have embeddings that represent strong visual features that show different outputs.

The embedding models used represent various types of image feature embedding methods. For example, Resnet50 is used as a baseline for the exploration to identify cluster outputs and find an optimal (min)cluster size and silhouette score for comparison. This enables a general cluster size and silhouette score to strive to or have an idea of where CLIP and BLIP-2 embedding models should fall. CLIP uses text and visual training data while BLIP-2 just uses visual training data.

## 2.2 CLUSTERING WITH HDBSCAN

HDBSCAN is the density-based clustering method used in the research to identify the best representation of actual topical clusters within the image corpus. The HDBSCAN algorithm (Hierarchical Density Based Spatial Clustering of Applications with Noise), is a clustering algorithm that groups features using hierarchical partitioning to maximize the stability of the selected clusters while also accounting for noise, and works well with real-world applications of imbalanced data (Campello et al., 2013). For example, the primary parameter is minimum cluster size, which sets a minimum size of a cluster. If a minimum cluster is too large it can group smaller clusters together and even include noise. If a minimum cluster size is too small it can include noise as a cluster and make unnecessary groups. Here, various min_cluster sizes were compared by identifying the number of clusters in the group and evaluating the clustering effects using the silhouette score. In this case, the silhouette score helps to identify an optimal cluster number while tuning the clustering algorithm. To get a better understanding of the clusters and to see how HDBSCAN functions, ResNet50 is used as a way to tune parameters and the outcome is shown in Appendix 1.

Clusters are labeled with numbers starting at 0, if a point is labeled with a -1, it means it has no cluster and can be identified as noise, or part of an outlier group. When calculating the silhouette score, points with the label -1 are filtered out to identify true cluster metrics. Parameter tuning showed that smaller min_cluster sizes produced more clusters (9–11), while larger values produced fewer clusters with higher silhouette scores, indicating trade-offs between nuance and separation. Because this algorithm is based on a cluster's density, there might be too many similar clusters in one area to discern specific boundaries. If we compare this score to the lowest silhouette score, 0.25 score with 9 clusters identified for a minimum cluster size of 30, there are more clusters identified. This lower score with higher clusters compared to the highest score with less clusters shows that there might be some overlapping clusters within highly dense areas of the image feature embeddings.

The clusters found are processed to show their density at a 2D level, and the file paths are included and listed for each cluster within each model. This process will provide a better understanding of what each cluster represents and how large these clusters are. It will include similar evaluation metrics and Human-in-the-Loop reviews of samples from each cluster based on content analysis methods.

## 2.3 EVALUATING ALL MODEL EMBEDDINGS

Three evaluation metrics will be used, to compare both parameters, and how they affect the different number of clusters. These metrics include the Silhouette Score, Calinski-Harabasz Index, or the Variance Ratio Criterion, and the Davies Bouldin Score. The Calinski-Harabasz Index (CH) quantifies the ratio of intracluster and intercluster variance, i.e. the ratio of within and between cluster variance (Rachwał et al., 2023). A higher CH index shows that clusters are dense and well separated, but there is no index score upper bound, making the score interpretation relative to other metrics (Rachwał et al., 2023). On the other hand, the Davies-Bouldin index (DBI) method focuses on the average similarity ratio between each cluster and its most similar cluster, not just all clusters (Ashari et al., 2022; Rachwał et al., 2023). A smaller DBI score shows an improvement in the clustering output for the clustering algorithm (Ashari et al., 2022; Rachwał et al., 2023). The clustering output for each model can be evaluated using the same metrics, but the embeddings will be different because they will have gone through a dimensionality reduction with the UMAP process. UMAP, or Uniform Manifold Approximation and Projection, is a dimensionality reduction technique that visualizes embeddings (image or text) in a high dimensional space, using its default parameters (McInnes et al., 2018). The results show that reducing dimensionality from image embeddings is probably not ideal for clustering the data into real groups, but it is preferable to visualize with the human eye. These low-dimensional features are run through the HDBSCAN algorithm with the set parameters found in the previous tuning process for each of the models. The clusters found here are visualized in 2D space CLIP in Figure 1 and ResNet50, and BLIP-2 in Appendix 2, and a comparison of their evaluation metrics with the tuned parameters is shown in Table 1.

The evaluation metrics of the tuned embeddings for the cluster densities still show a low silhouette score albeit improved, but have slightly higher CH Indices and lower DBI scores. In this case, even though the intracluster scores are low, the cluster separation or boundary scores are relatively decent. Here we can assume that the clusters are valid but most likely overlap each other.

Table 1: Minimum Cluster Size Parameter Tuning on ResNet50 Embeddings

| Tuned Evaluation Metric Parameters | ResNet50 | CLIP | BLIP-2 |
|---|---|---|---|
| Silhouette Score | -0.36 | -0.25 | -0.327 |
| Calinski Harabasz Index Score | 7715.64 | 14801.54 | 5612.30 |
| Davies Bouldin Index Score | 0.25 | 0.29 | 0.42 |

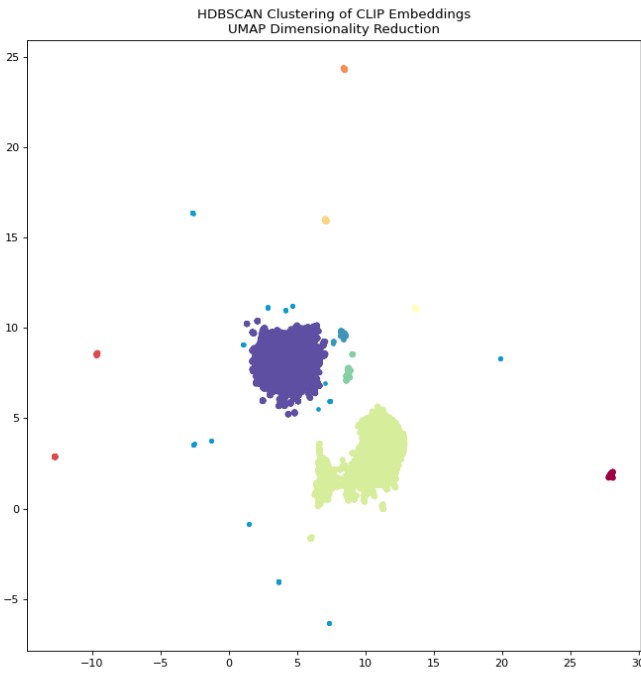

Figure 1: Visualization of UMAP clusters on CLIP embeddings with tuned parameters, Nearest Neighbor = 25, Minimum Distance = .01

There are typically two larger clusters with a few smaller clusters surrounding the larger groups for each model's embeddings. These visuals show a separation for two types of groupings, except for BLIP-2, with potential room to split these clusters into different classifications. These results also show that there is still a high density while clustering these image embeddings, meaning that there are images with nuance that cannot be identified with just feature embeddings, and that noise might be more influential than expected, particularly when dealing with outliers.

## 2.4 CLUSTER CONFIDENCE

Aside from density, we can get an overall look at the outliers and confidence scores within the clusters for each feature embedding model. The HDBSCAN clustering algorithm from above provided different attribute outputs for the data, including point probabilities and a cluster label. The point probability output reflects the probability that a point, or image, actually belongs to a specific cluster, the higher the probability, the more likely that point belongs to that cluster. The cluster labels are a number label that relates to a specific cluster. A high confidence score for a point within a cluster is above 0.85 is the set threshold because it represents more sure, or confident feature embeddings within the model. A look at the average high confidence score provided by each model, and the percentage of points with a confidence score above the 0.85 threshold can be found in Appendix 3.

As each model provided a different number of clusters, these clusters also varied in sizes. For example, as the ResNet50 model identified 7 clusters, the intuition here is that there are clusters with more points than there are smaller in size. The same can be said for the CLIP model that found 9 clusters. The BLIP-2 model identified 28 clusters, potentially identifying smaller clusters that could have accounted for more nuanced images. Many of the clusters have points with a confidence above 0.7, potentially indicating that each model had a decent amount of training data that supports the image data. ResNet50 has much of its data between 0.75 and 1.0 confidence levels, with very few around 0.6. The CLIP model has points that mostly hover between 0.8 and 1.0, but does show points with the lowest confidence. BLIP-2 is also skewed between 0.7 and 1.0, and has a larger spread between these confidence points; it also has a small portion of its point probabilities around 0.6, see distribution in Appendix 2. These results also show that small semantic nuances might provide a more robust representation of the data and how it supports the overall narrative structure. A content analysis and annotations of sampled data from each cluster of the three models is carried out in the next section to evaluate the clustering quality and compare the output to the evaluation metrics provided here.

## 3 HUMAN IN THE LOOP EVALUATION: INDUCTIVE CONTENT ANALYSIS AND ANNOTATION

This work implements a shared control of interactive machine learning within the realm of Human in the Loop (HITL) Evaluation (Amershi et al., 2014), where the human and the learning system are iteratively supplying focused information by evaluating the different clusters, and in the upcoming sections, different subgroups of the network. Interactive machine learning has been used specifically for image classification, image segmentation, time series data, text data, and complex data at scale. At first, specific frames were explored from reading through the literature. These frames influenced the ideation of different label classifications from a first round annotation of 50 images. Initial labels included protest, solidarity, selfies, banners, and images with text, later refined through iterative annotation. A description can be found in Appendix 4.

### 3.1 INDUCTIVE CONTENT ANALYSIS: A SYSTEMATIC APPROACH

Images representing the meaning of a social movement's messaging as a whole have the ability to provide a type of knowledge useful to the social movement that enables an understanding of the dynamics that encompass its grievances (Kuhn, 1985). For example an image post on social media has the ability to surpass large media blackouts and provide support from different perspectives (Martínez Rod & Mitchell, 2025). Different perspectives can support or reject the ideology of the social movement and be spread across the platform with contrasting meanings to specific ideological groups (Ribeiro et al., 2018). This practice can lead to various types of bias that contextualize an image based on its supporting information. Different types of bias like image content bias, contextual information bias, and platform bias based on the affordances. The contextual information bias might come from the image's metadata, and the image content bias would come from the different objects or iconography that make-up the image; or even the topics covered within the text that is included in the image (Lin et al., 2021). Platform bias deals with the affordances of the Instagram algorithm that allow for specific types of images formatting and the number of images that can be posted. Regardless, knowing and identifying these types of biases provides stronger contextual support for analyzing the overall network structure of the data.

Organizing social movement knowledge as interdependent activities that represent a multitude of various frames, within clusters has the ability to identify topics and themes within the content, and explore similarities. Inductive content analysis groups data together through the process of abstraction to answer a research question using concepts, categories, or themes on the unit of analysis (Kyngäs, 2019). Here, codes are identified to organize the data into various categories, concepts and themes, this process of abstraction is done iteratively to show how the data is connected within each cluster (Kyngäs, 2019). In this research, the inductive coding process is implemented by using a systematic approach. The systematic content analysis in this work is considered a qualitative approach to build knowledge within the research (Finfgeld-Connett, 2014). It calls on the image's collection of the metadata, image id, main hashtag, alternative text provided by Instagram, the account poster, the number of likes, the accompanying post comment, and the data it was posted.

The sample images of 185 were chosen from the 6 clusters found in ResNet50, the 8 clusters found in CLIP, and the 27 clusters found in BLIP-2. These images are selected based on their average probability score identified by the HDBSCAN model. Ten maximum and ten minimum of these image probabilities were taken from each cluster and the top image was analyzed using this coding process for the three models. An example of the model used, image name, cluster label, probability score, outlier score, and where the image lands in the top or bottom samples pulled for each cluster can be seen in Appendix 5. These probabilities are extended to include the content analysis and annotation coding schema.

## 3.2 CONTENT ANALYSIS: REVIEWS FROM IMAGE CODING

Getting a review of the sentiment for the different cluster groups is done by using the combination of the probability columns and the content analysis and annotation columns to better understand the narrative messaging of the cluster. Each model provided different clusters which represented different types of hashtag and account combinations of images. Because the images were collected based on hashtag and account search, the content analysis will be implemented on the different hashtag mentions within and across the model's clusters. A review of each model's cluster hashtags and accounts is carried out in the inductive content analysis. It is important to note that if a post had a carousel of images, or multiple images to swipe through, the original account comment and hashtags remain the same for each image in that post. In other words, multiple images can have the same hashtag and original account comment. Typically these images relate to the same topic and hold similar styles throughout.

### 3.2.1 RESNET50 IMAGES REPRESENTING EACH CLUSTER

ResNet50 clusters included hashtags such as #8M, #FuimosTodas, and #UnDiaSinNosotras, which often overlapped across models, reflecting topical density. A detailed account of each hashtag use across clusters can be found in Appendix 6. The different images that come with the hashtag #violenciadegenero (gender violence) show this wide range of content across clusters. Similarly, #8M stands for the 8th of March, which is international women's day; a day that sparked a lot of the anti-feminicide movement's messaging and call to action. This hashtag is also a broad reach across women's issues that can represent various topics relating to international woman's day. For example, the top row of Figure 2 shows two very different styles and image content that touches on the anti-feminicide movement, and content that comments on other types of women's issues; granted these images represent the outlier group that is a stand-in for noise in the data. Here, the model seems to identify centered text on a plane background as noise, according to the images identified to represent the cluster.

Across the clusters, the image content touches on or represents different perspectives of the anti-feminicide movement, along with few for the women's movement. Some of these images are informational in the sense that they provide definitions, examples of media support, or advice surrounding feminicide as in clusters 4, 5, and 6. Some of these images might be biased because of potential media influence and group advertisement. Other representative images are accusatory and highlight comparisons of women and men, such as the text in clusters 1 and 2. These show some bias through one-sided emotional personal experiences, and book promotion. Other image clusters represent groups in a positive and a supportive light to provide solidarity for anti-feminicide, where other groups are shown in the middle of protest with a more aggressive activism tone, seen in clusters 0 and 3. The bias potential here comes from the first hand participant sharing along with a digital flyer created by a group to show a happier supportive light.

### 3.2.2 CLIP IMAGES REPRESENTING EACH CLUSTER

CLIP clusters included hashtags such as #yotecreo, #undiasinnosotras, brujamixteca, #violencia-machista, #violenciadegenero, #FuimosTodas. These hashtags and accounts were not used in each cluster, but across clusters. A detailed look at where each was used, along with the size of each cluster can be found in Appendix 7. Looking at the cluster output, #8M is found in three clusters, -1, 5, and 8, similar to the ResNet50 model. This is understandable because it represents the 8th of March, international women's day. Similarly, #UnDiaSinNosotras (A day without us) is a hashtag representing the day after #8M where Mexican women take to the streets to strike, or just 'dissapear'

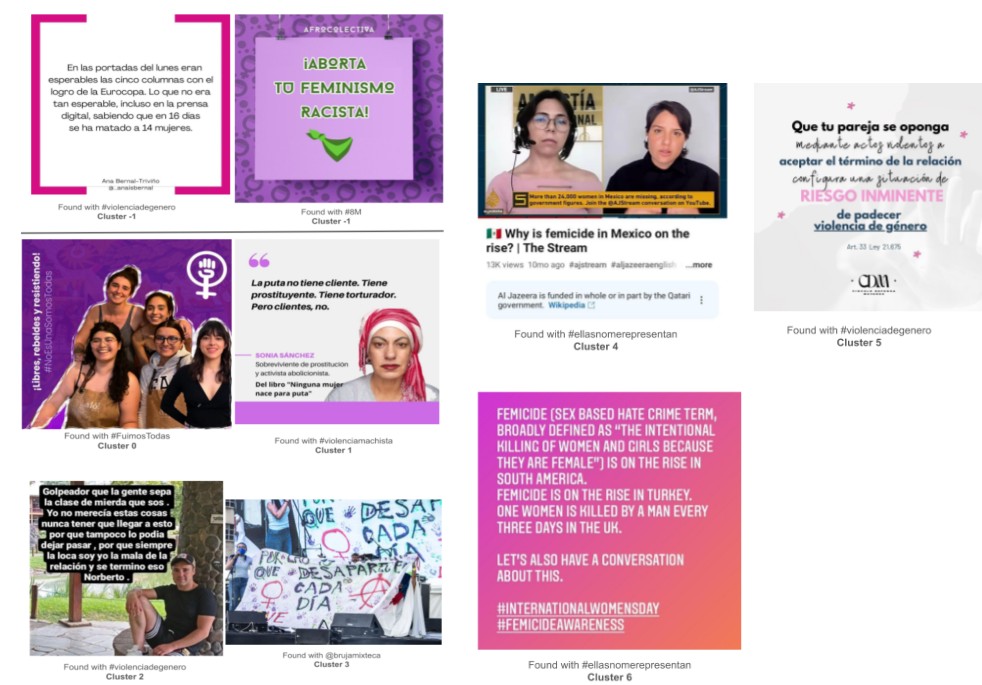

Figure 2: Images representing the clusters and outlier groups for ResNet50

from their daily life to protest gender violence and feminicide (Semple & Villegas, 2020). Another hashtag found throughout CLIP's clusters is #FuimosTodas (It was all of us) which is a common hashtag used in the feminist movement in Mexico. It is an interesting hashtag because it was initially used as a feminist hashtag in solidarity and was redefined within the context of feminism and the Mexican state (Morocho & Delgado, 2024). The images in Figure 3 represent the 8 clusters, and the outlier group found by CLIP.

The CLIP clusters span different perspectives of the anti-feminicide movement in that they touch on more specific narratives of the movement. For example, based on the hashtag #25N, cluster 5's image is in support of the elimination of violence against women in a symbolic manner, whereas cluster 3 and cluster 4 use the hashtag #FuimosTodas (It was all of us) and show a larger type of support being flown around in a blip, providing more information and ideological phrases that represent the anti-feminicide movement. Cluster 0 shows an image of a protest from a group account, brujamixteca, that actively shows support for the movement. Cluster 6 exemplifies the comments and back-and-forth done online to continue supporting specific days such as March, 9th, or *Un dia sin nosotras* (a day without us) where women attempt to stop their daily routine and show how their absence might be missed. Some tangential topics were identified with Clusters 1 and 2, again, they review the topic of gender or domestic violence; and cluster 8 deals with the need for sexual education. Cluster -1, the outlier group, shows an image of only text on a plain background, similar to the outlier group content of ResNet50. This is potentially indicative of the models not being able to 'recognize' the text as real words and understand their sentiment. Other than clusters 3 and 4 which have obvious overlap, the sampled cluster representations are visibly different.

Some bias was shown here with different emotional tones such as aggressiveness, somber, solidarity, and a type of ominous representation (particularly with the blimp aircraft images). These emotional tones show bias as they attempt to bring about some type of feelings of outrage or pity for victims of the cause. These image representations show little informational content and rely on protest or solidarity types of framing within the messaging.

### 3.2.3 BLIP-2 Images Representing each Cluster

The BLIP-2 model found 28 clusters plus the outlier group. Here, 20 of the hashtags and accounts used to search for the image posts were found in the annotations. Some of these hashtags and ac-

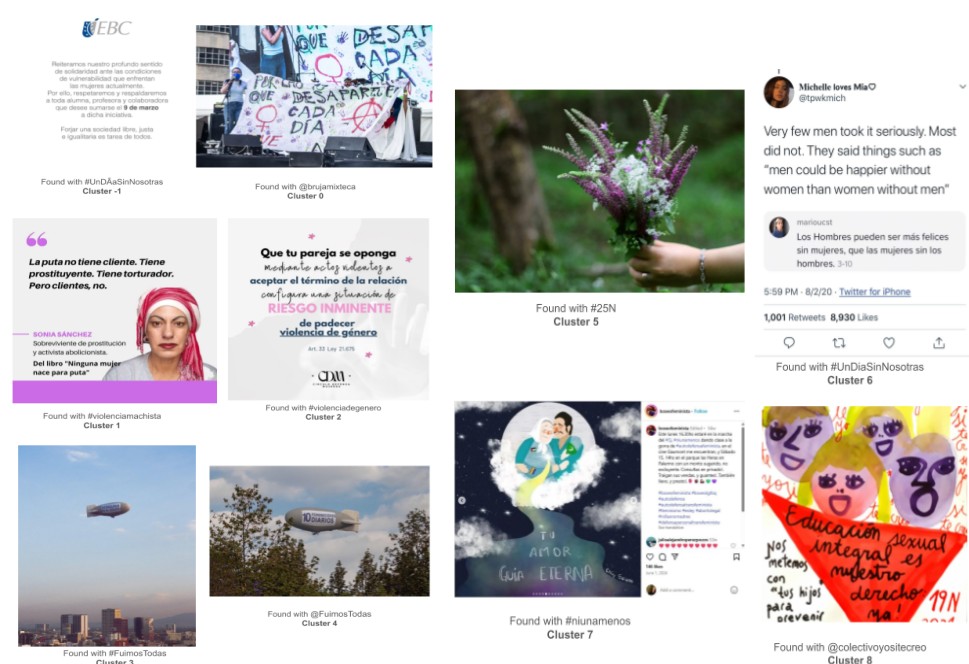

Figure 3: Images representing the clusters and outlier groups for CLIP Model

counts include, #violenciadegenero, antimonumenta_vivasnos, #UndiaSinNosotras, #noestamosto-das, #nonoscallamosmas, #ellasnomerepresentan, #femicideinmexico, #yotecreo, abogadafemina. A detailed look at where each was used, along with the size of each cluster can be found in Appendix 8. This table is particularly important here as it might show instances of cluster overlap in relation to the representative images found in Figure 4.

These image cluster representations show how the visual overlap across the cluster groups is apparent. For example, clusters 7, 12, 20, 21 have long or multiple text blocks in a type of information of memo style. Clusters 8, 9, 19, 24, 25 have a similar format of purple background with an image and text overlaid. The multiple clusters found enable a closer look at various themes represented in the images and their text. Some of the content covers actual feminicide stories or reports, where other images discuss the support or solidarity for protests, particularly for a day without a woman protest and international women's day. The account abogadafemina had an image representation across many of the clusters with overlapping visual content styles (clusters 19, 24, 25), but also some cluster representations that were standalone in their image (cluster 14). There was some bias shown as advertisements were using the "day without a woman" march to promote their business, and it was showing how universities were sending memos in support of the march. Topics of gender violence and specific feminicide stories were also sprinkled in the representative images, providing examples of the various ways they can be presented; clusters 9 and 21 for example.

## 4 DISCUSSION

Overall, the clusters show that the data is densely packed, meaning there is a lot of visual overlap across the collection of images. This is not surprising as the images are topically related so their content will also be similar. Many of the images contained text which enabled a better reading or analysis of the image while implementing content analysis. The topics found in the clusters across the three models included, woman/male comparisons, accusers, life examples, domestic violence, gender violence, protest phrasing, support or in solidarity of someone or a cause. The models were able to show a range of the topics by clustering the images into different group sizes.

True clusters have been defined as context dependent (Hennig, 2015). Because many of these clusters show very dense, or overlapping clusters based on the normalized image features and human

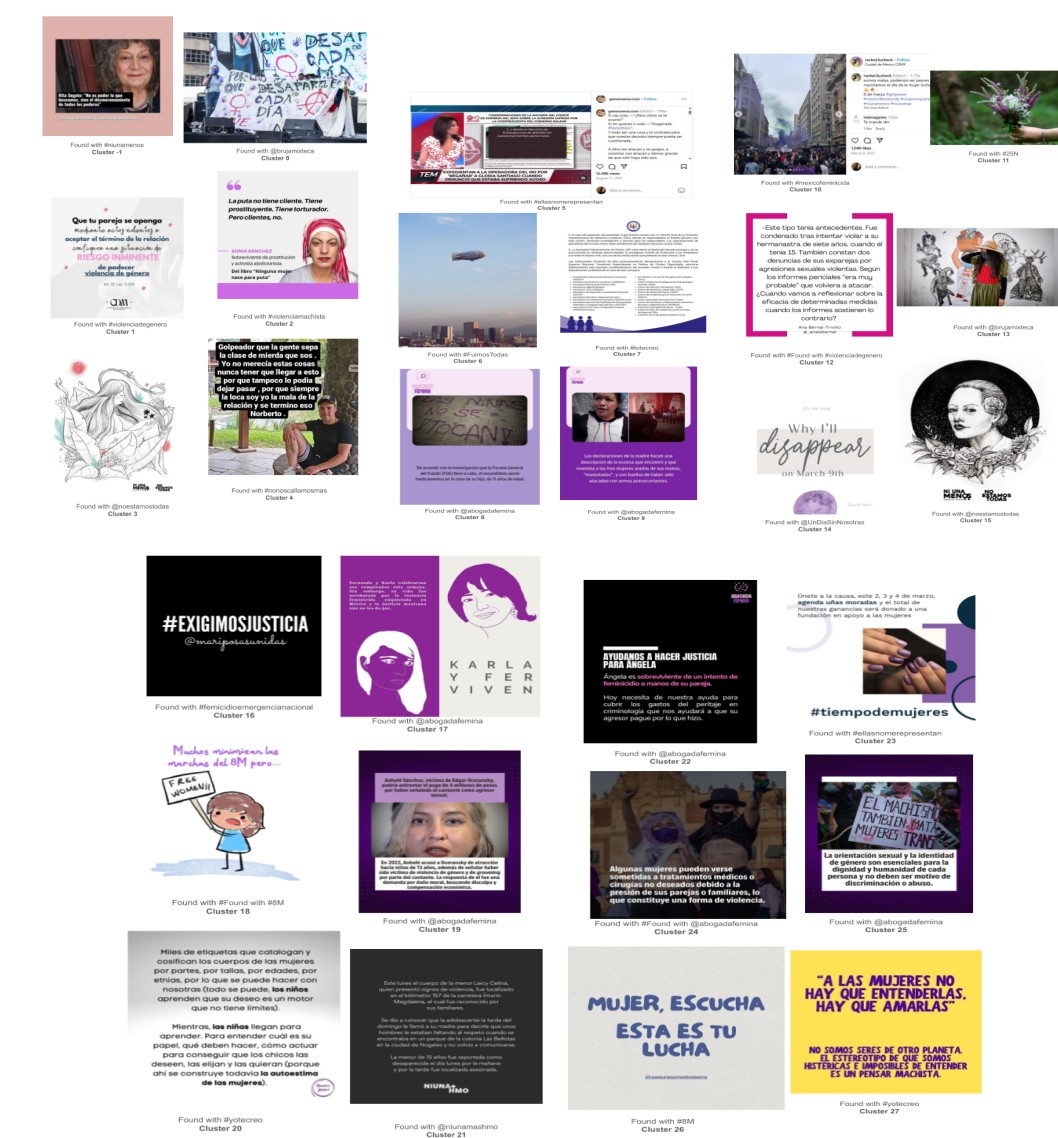

Figure 4: Images representing the clusters and outlier groups for the BLIP-2 Model, continued

in the loop analysis, future work could be to label the images based on topical tags that represent a sample of the corpus. The best separation results from the CLIP model suggest that 9 clusters are the best cluster output. The CLIP model showed to have the best Silhouette Score (-0.25) and Calinski-Harabasz Index Score (14801.54). Although these clustering metrics are high, a negative silhouette score might indicate that there is misclassification by the model. Regardless, this model overall had the best evaluation metrics, and the qualitative content analysis showed a distinct range of topics across the clusters. With this finding, it is also important to note that the clusters show overlap, found both in the quantitative and qualitative evaluation. The overlap can be argued that there is nuance within the image feature embeddings.

Future work for this research will create labels and annotate the data to identify clearer groups. Specifically, working with multimodal LLMs to label the data and compare that against a sample of human-annotated data could be a strong way to create clear groupings and analyze how each image is related to another.These relations could be analyzed using semantic similarities connected in a graph structure, including more human in the loop evaluation. Instagram policies do not enable open sharing of the content. To make this work reproducible, creating synthetic data to test this process for comparison and as a baseline for customized collections of topical images.

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

APPENDIX

Table A1: Appendix 1; Minimum Cluster Size Parameter Tuning on ResNet50 Embeddings

| Minimum Cluster Size Parameter | Silhouette Score | Number of Clusters |
|---|---|---|
| 20 | 0.236 | 11 |
| 30 | 0.254 | 9 |
| **40** | **0.621** | **6** |
| 50 | 0.522 | 5 |
| 60 | 0.529 | 4 |
| 70 | 0.540 | 4 |
| 80 | 0.565 | 3 |
| 90 | 0.561 | 3 |

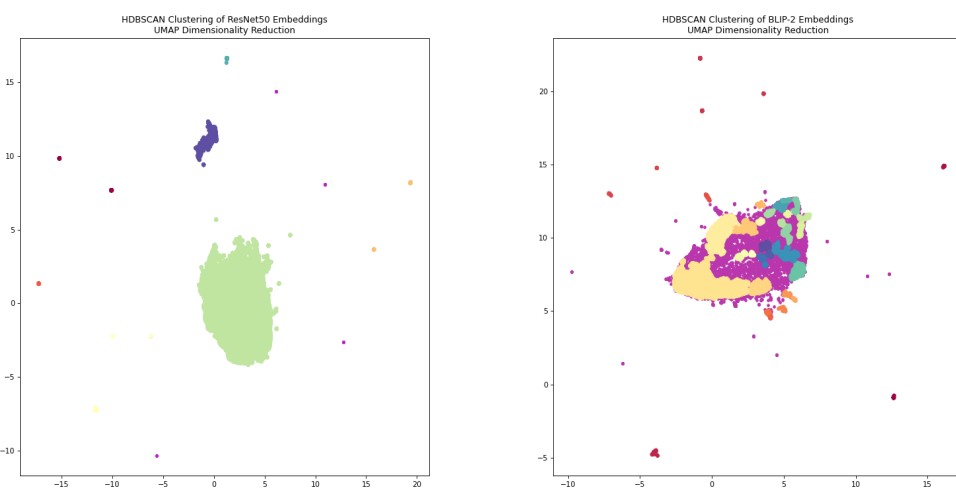

Figure A1: Appendix 2: Visualization of UMAP clusters on ResNet50 and CLIP-2 embeddings with tuned parameters, Nearest Neighbor = 25, Minimum Distance = .01

Table A2: Appendix 3: Review of High Confidence Points per Model

| | ResNet50 | CLIP | BLIP-2 |
|---|---|---|---|
| High Confidence Point Probability % Found Above the Threshold | 96% (0.96) | 94% (0.94) | 67% (0.67) |
| Avg. High Confidence Point Probability Across Clusters | 0.99 | 0.99 | 0.98 |
| Number of Clusters Found | 7 | 9 | 28 |

Appendix 4: These class labels included 'a photo of a protest', 'an image of solidarity', 'an image of an illustration', 'a photo of a group', 'photo of a sign(s) or banner', 'a selfie', 'an image with text in Spanish', 'an image with text in English', 'a photo of landmarks or buildings'. These labels were used as a starting point for implementing a content review and annotation pass of the images. When qualitatively checking a handful of random images from each cluster, image duplicates that were posted by different accounts, likely reshares, would dominate the cluster's top probabilities, which is understandable. But, what was also interesting is that topics not related to the anti-feminicide

movement, like #25N, a type of airplane, would get its own cluster. This led to an editing of the class labels to re-run the models, and a re-paramerization of the HDBSCAN clustering algorithm.

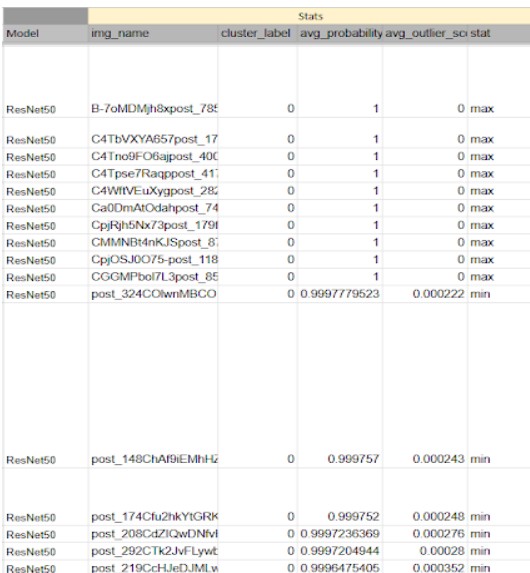

Figure A2: Appendix 5: Example of max and min probabilities of first ResNet50 cluster

Table A3: Appendix 6 - Hashtag and Account Usage per ResNet50 Cluster. Size of cluster is also noted.

| Cluster | Size (image points) | Hashtag/Account (Total annotated in cluster) |
|---|---|---|
| −1 (Outliers) | 53 | #8M (4); #Violenciadegenero (3) |
| 0 | 67 | #FuimosTodas (4); #noestamostodas (3) |
| 1 | 56 | #violenciamachista (7) |
| 2 | 43 | #noestamostodas (2); #nonoscallamosmas (5) |
| 3 | 243 | @abogadafemina (1); @brujamixteca (5) |
| 4 | 14,742 | #8M (1); @colectivoyositecreo (1); #ellasnomerepresentan (1); #violenciadegenero (2) |
| 5 | 81 | #nonoscallamosmas (1); #violenciadegenero (4) |
| 6 | 1,283 | @antimonumenta_vivasnosqueremos (1); #ellasnomerepresentan (1); @niunamashmo (2); #violenciadegenero (2) |

Table A4: Appendix 7 - Hashtag and Account Usage per CLIP Cluster. Size of cluster is also noted.

| Cluster | Size (image points) | Hashtag/Account (Total annotated in cluster) |
|---|---|---|
| −1 | 177 | #8M (1); #UnDiaSinNosotras (2); #niunamenos (1); #yotecreo (1) |
| 0 | 232 | @brujamixteca (4); #violenciamachista (1) |
| 1 | 89 | #nonoscallamosmas (2); #violenciamachista (2) |
| 2 | 76 | #violenciadegenero (5) |
| 3 | 108 | #FuimosTodas (5) |
| 4 | 43 | #FuimosTodas (5) |
| 5 | 7,288 | #25N (3); #8M (1); #FuimosTodas (1) |
| 6 | 111 | #UnDiaSinNosotras (3); #ellasnomerepresentan (1) |

| Cluster | Size (image points) | Hashtag/Account (Total annotated in cluster) |
|---|---|---|
| 7 | 182 | #ellasnomerepresentan (2); #niunamenos (1); @nos_hacen_falta (1) |
| 8 | 8,261 | #8M (2); #UnDiaSinNosotras (1); @colectivoyositecreo (1); @niunamenoshmo (1) |

Table A5: Appendix 8 - Hashtag and Account Usage per BLIP-2 Cluster. Size of cluster is also noted.

| Cluster | Size (image points) | Hashtag/Account (Total annotated in cluster) |
|---|---|---|
| −1 | 2,898 | #8M (1); #niunamenos (1); #violenciamachista (1) |
| 0 | 244 | #FuimosTodas (1); @brujamixteca (2) |
| 1 | 77 | @antimonumenta_vivasnosqueremos (1); violenciadegenero (2) |
| 2 | 69 | UnDiaSinNosotras (1); violenciamachista (2) |
| 3 | 49 | @noestamostodas (3) |
| 4 | 61 | @noestamostodas (1); #nonoscallamosmas (2) |
| 5 | 67 | #ellasnomerepresentan (1); #femicideinmexico (1); @nos_hacen_falta (1) |
| 6 | 111 | #FuimosTodas (3) |
| 7 | 87 | #FuimosTodas (1); #nonoscallamosmas (1); #yotecreo (1) |
| 8 | 115 | @abogadafemina (3); @noestamostodas (1) |
| 9 | 92 | @abogadafemina (3) |
| 10 | 114 | #ellasnomerepresentan (1); mexicofeminicida (2) |
| 11 | 8,628 | 25N (1); 8M (1); abogadafemina (1) |
| 12 | 601 | abogadafemina (1); violenciadegenero (1); violenciamachista (1) |
| 13 | 199 | FuimosTodas (2); @brujamixteca (1); #yotecreo (1) |
| 14 | 50 | 8M (1); UnDiaSinNosotras (1); abogadafemina (1) |
| 15 | 60 | 8M (1); FuimosTodas (1); noestamostodas (1) |
| 16 | 72 | femicidioemergencianacional (1); noestamostodas (2) |
| 17 | 50 | FuimosTodas (1); abogadafemina (1); noestamostodas (1) |
| 18 | 153 | 8M (2); noestamostodas (1) |
| 19 | 1,060 | abogadafemina (2); nos_hacen_falta (1) |
| 20 | 44 | nonoscallamosmas (1); #yotecreo (2) |
| 21 | 45 | abogadafemina (1); niunamashmo (2) |
| 22 | 162 | abogadafemina (3) |
| 23 | 256 | #ellasnomerepresentan (1); noestamostodas (1); #yotecreo (1) |
| 24 | 88 | abogadafemina (2); violenciamachista (1) |
| 25 | 127 | abogadafemina (3) |
| 26 | 486 | 8M (2); vivasnosqueremos (1) |
| 27 | 502 | violenciadegenero (1); #yotecreo (2) |

