# OpenReview forum: "Image Embeddings from Social Media: Computer Vision and Human in the Loop Applications for Social Movement Messaging"
_ICLR.cc/2026/Conference — Submitted to ICLR 2026_

### Official Review · Reviewer_2Fmx · 2025-10-25

**Soundness:** 1
**Presentation:** 2
**Contribution:** 1
**Rating:** 2
**Confidence:** 4

**Summary:**

This paper deals with the problem of analyzing images shared on social media platforms. More specifically, images related to a specific topic are collected from a single social media platform, and image features extracted from standard image encoders are grouped using DBSCAN, a standard clustering algorithm in data mining and database management. Extracted clusters are checked and investigated by human inspections, which reveal that different image encoders yield different clusters and thus provide different semantic groups.

**Strengths:**

S1. The research topic dealt with in this paper is significant. Social media has become one of the most influential media platforms, and its significance continues to grow day by day. In this sense, analyzing the characteristics and dynamics of media content distributed on social media platforms is one of the most significant research topics for understanding the shifts in social conditions and public opinion.

**Weaknesses:**

W1. If my understanding is correct, the main topic of this paper belongs to social science, not computer science. In this sense, I strongly recommend this paper to be submitted to other conferences related to social science, such as ICWSM and CHI. ICLR focuses on fundamental theories and innovative technologies for machine learning, placing a high priority on theoretical and/or technical novelty.

W2. On the other hand, discovering and demonstrating novel findings with already known techniques is also valuable for healthy development in computer science. However, papers focusing on this aspect should provide extensive investigations from various viewpoints and attempt to address nearly all questions derived from the original research question and the experimental results. See e.g. [Teney+ CVPR2024 https://openaccess.thecvf.com/content/CVPR2024/html/Teney_Neural_Redshift_Random_Networks_are_not_Random_Functions_CVPR_2024_paper.html]. From this viewpoint, the current paper seriously lacks deep investigations for the research question.

W3. The organization should also be majorly revised. For example, this paper devotes excessive space to explaining well-known techniques. Meanwhile, almost all the experimental results required for describing the main story are placed in the supplementary material.

**Questions:**

Q1. I could not understand the reasons why the authors chose the anti-feminicide movement as research material. This description is required for understanding the philosophy of this paper and the research question of this paper and checking the existence of ethical issues.

---

### Official Review · Reviewer_3vfF · 2025-10-31

**Soundness:** 1
**Presentation:** 2
**Contribution:** 1
**Rating:** 0
**Confidence:** 5

**Summary:**

6,567 image posts from Instagram related to the anti-feminicide movement in Mexico were collected and analyzed to see if these models can group the pictures into meaningful topics (like protest signs, solidarity posts, info posters). Human oversight is also provided to make sense of such groups or clusters.

**Strengths:**

- The social problem is very relevant as we need to study what is going on in our society and how anti-feminicide discourse is prevalent online.
- Above 16,000 image posts from Instagram are taken and studied, which is a big number.
- The usage of multiple vision algorithms like (ResNet50, CLIP, BLIP-2) and multiple clustering methods gives both depth and breadth to the study.
- There is a good attempt to connect automated clustering with human oversight which helps sensemaking of clusters effective.
- Detailed discussion like acknowledgement of challenges of dense-clusters and text-rich images.

**Weaknesses:**

- The methodological contribution is very minimal. The paper applies a few algorithms with clustering techniques. There is no new method or any novel grounding on the modeling side. Evaluation is very descriptive rather than quantitative.
- The clustering gives negative silhouette scores, and the authors claim highly useful clustering performance.
- There is no comparison to anything. Like other clustering algorithms (k-means?). There is no usage of multimodal pretrained models that are predominantly tuned in social media data.
- The work is very exploratory. There is no hypothesis-driven research. There is not much scientific takeaway from the paper for ICLR audience.
- The paper has overemphasized the social sciences. It may be valuable socially as it tackles important social problems but for a machine-learning venue, the contribution is so limited.

**Questions:**

- Why did you conclude "best separation" when all silhouette scores remain negative?
- Did you try any OCR + text embeddings? The images have very dense text. While CLIP/BLIP could handle text + image, did you not try posing the problem differently and aim for a better representation?
- Why did you use only a few clustering approaches?

---

### Official Review · Reviewer_JWhX · 2025-11-02

**Soundness:** 2
**Presentation:** 2
**Contribution:** 2
**Rating:** 2
**Confidence:** 2

**Summary:**

This paper analyzes 16,567 Instagram images from the anti-feminicide movement in Mexico using unsupervised and self-supervised embedding models combined with HDBSCAN clustering. The authors employ human-in-the-loop content analysis to evaluate cluster quality and understand visual messaging structures. The results show dense, overlapping clusters across all models, with CLIP achieving the best separation metrics.

**Strengths:**

- The paper provides a thorough comparison of three embedding approaches and combines quantitative clustering metrics with qualitative human annotation of 185 sample images.
- The application to anti-feminicide social movement messaging represents a meaningful use case for computer vision methods. The dataset of 16,567 Instagram posts provides a substantial corpus, and the human-in-the-loop analysis reveals real evaluations that the VLMs cannot find.

**Weaknesses:**

- The paper applies existing, well-established methods in an off-the-shelf manner. There are no new proposals for model development.

**Questions:**

- How do you justify claiming CLIP is "best" when DBI of CLIP is worse than that of ResNet50 or all the methods showed the negative Silhouette scores?

---

### Official Review · Reviewer_9LwA · 2025-11-02

**Soundness:** 2
**Presentation:** 2
**Contribution:** 2
**Rating:** 2
**Confidence:** 4

**Summary:**

This paper investigates the use of computer vision-based image embeddings and clustering for analyzing social movement messaging within a large set (16,567 posts) of Instagram images related to the anti-feminicide movement in Mexico. The study extracts feature embeddings using ResNet50, CLIP, and BLIP-2, applies HDBSCAN for clustering, and evaluates clusters with several quantitative metrics alongside human-in-the-loop inductive content analysis. The results compare representational properties across models and discuss overlap and nuance in image messaging structure, highlighting the strengths and limitations of current representation models for domain-specific, topic-coherent social images.

**Strengths:**

1. The paper addresses a relevant, underexplored problem at the intersection of machine learning, social movements, and computational social science, providing quantitative insight into the structure of visual messaging in a humanitarian context.

2. It adopts a comparative framework, using both popular (ResNet50, CLIP) and more advanced (BLIP-2) image embedding models, allowing a nuanced analysis of their clustering behavior on real-world activist imagery.

3. Employing multiple established clustering evaluation metrics (Silhouette Score, Calinski-Harabasz Index, Davies-Bouldin Index), in tandem with human-in-the-loop content analysis, demonstrates methodological rigor and brings valuable qualitative depth to quantitative findings.

**Weaknesses:**

1. While ResNet50, CLIP, and BLIP-2 are established models, the rationale for selecting these, particularly why not use more recent or task-specialized models (e.g., multimodal sentiment/abusive meme detectors), is underspecified. There is also little reflection on how text-in-image (e.g., hashtags, slogans) is handled beyond embedding, although text is central to social movement images.

2. No non-deep-learning clustering baselines (e.g., classical SIFT/ORB features, PCA+GMM, or manual curation) are reported for context. Similarly, the study provides limited statistical testing to gauge whether any performance differences (or cluster count/size differences in Tables/Appendices) are meaningful, or merely artifacts of parameter tuning.

3. While the analysis (Section 3 and Figures 2–4) is a valuable complement, the use of coding to identify cluster validity serves more as a post hoc rationalization than a systematic validation, potentially overfitting the interpretation to noisy clusters. For instance, the claimed “nuance” within dense groups could mask model or clustering failures. More rigorous validation (possibly co-clustering, cross-validation with held-out hand-labels, or even crowd-sourced validation as secondary annotation) is missing.

4. Given the consistently negative Silhouette Scores (Table 1), what measures (quantitative or qualitative) can you provide to justify that your clusters are not artifacts of parameter tuning, but capture semantically meaningful differences? Would alternative clustering/objective functions mitigate the issue of dense overlap?

5. Can you provide any ablations or error analysis comparing embeddings and clusters for images dominated by text versus those that are mostly visual? How do ResNet50, CLIP, and BLIP-2 differ in treating such cases?

**Questions:**

As shown in Weakness.

**Details Of Ethics Concerns:**

1.  The proposed dataset includes 16,567 Instagram images from users discussing the anti-feminicide movement in Mexico. The paper does not specify whether the images were collected under Instagram’s terms of service or whether users’ consent was obtained for data scraping. Since Instagram content may include identifiable individuals, victims, and activists, this raises privacy and potential safety concerns.

2. The dataset revolves around gender violence and feminicide, involving victims of crime and social protestors. Clustering and visualization of such content might inadvertently expose sensitive identities or personal narratives without sufficient anonymization.
There is no clear indication that ethical clearance (IRB/ethics board approval) was obtained for analyzing such sensitive human-generated content.

---

### Meta-Review · Area_Chair_R8Fk · 2026-01-05

**Summary:**

This paper collected and analysed Instagram posted images associated with anti-feminicide movement in Mexico to better understand messaging patterns on the platform. Using unsupervised (ResNet50) and self-supervised (CLIP, BLIP-2) embeddings with density-based clustering (HDBSCAN) and human-in-the-loop content analysis, the authors identified overlapping clusters, reflecting nuanced topics including domestic violence, gender violence, and solidarity. The work combines quantitative embeddings with qualitative coding of image content and metadata to explore thematic structures.

**Reviewer Concerns:**

Reviewers raised several concerns including:

* Clustering quality is weak, with consistently negative silhouette scores. Regardless, the paper draws strong conclusions about meaningful clusters without providing convincing quantitative or statistical justification.

* Baseline coverage is insufficient (no classical or alternative clustering baselines, no OCR+text modeling despite text-heavy images, no task-specialized or social-media-tuned models).

* Evaluation lacks rigor, relying heavily on post hoc qualitative interpretation rather than systematic validation.

* Claims comparing representation quality across models are not well supported by the reported metrics.

Furthermore, ethics concerns regarding the scraping and analysis of sensitive data related to gender-based violence, with insufficient discussion of consent, privacy, or ethical approval were raised.

**Reviewer Scores:**

The paper received Rating 2 (reject) from 3 reviewers and a Rating of 0 from the fourth reviewer.

---

### Decision · Program_Chairs · 2026-01-26

Reject